# Two-Stage Multi-Objective Stochastic Model on Patient Transfer and Relief Distribution in Lockdown Area of COVID-19

**DOI:** 10.3390/ijerph20031765

**Published:** 2023-01-18

**Authors:** Shengjie Long, Dezhi Zhang, Shuangyan Li, Shuanglin Li

**Affiliations:** 1School of Traffic and Transportation Engineering, Central South University, Changsha 410075, China; 2College of Logistics and Transportation, Central South University of Forestry and Technology, Changsha 410004, China

**Keywords:** patient transfer, relief distribution, two-stage stochastic model, multi-objective optimization

## Abstract

The outbreak of an epidemic disease may cause a large number of infections and a slightly higher death rate. In response to epidemic disease, both patient transfer and relief distribution are significant to reduce corresponding damage. This study proposes a two-stage multi-objective stochastic model (TMS-PTRD) considering pre-pandemic preparedness measures and post-pandemic relief operations. The proposed model considers the following four objectives: the total number of untreated infected patients, the total transfer time, the overall cost, and the equity distribution of relief supplies. Before an outbreak, the locations of temporary relief distribution centers (TRDCs) and the inventory levels of established TRDCs should be determined. After an outbreak, the locations of temporary hospitals (THs), the locations of designated hospitals (DHs), the transfer plans for patients, and the relief distribution should be determined. To solve the TMS-PTRD model, we address an improved preference-inspired co-evolutionary algorithm named the PICEA-g-AKNN algorithm, which is embedded with a novel similarity distance and three different tailored evolutionary strategies. A real-world case study of Hunan of China and 18 test instances are randomly generated to evaluate the TMS-PTRD model. The finding shows that the PICEA-g-AKNN algorithm is better than some most widely used multi-objective algorithms.

## 1. Introduction

From ancient times to the present, epidemics have been threatening the lives of people. Typical outbreaks in recent years include coronavirus disease 2019 (COVID-19) [1], H1N1 influenza [2], and Ebola [3,4]. In 2020, 80,000 people were infected with COVID-19 in Wuhan, the most severely affected city in China. To control the spread of the COVID-19 disease, the urban area of Wuhan was effectively blocked; that is, people and vehicles could not contact the outside world without permission.

In a lockdown area, emergency management must transfer infected patients to hospitals and provide relief supplies to ensure infected patients receive timely medical treatment. Because of the sudden outbreak of an epidemic, lockdown areas are often unable to access relief supplies in time. Given the time required to replenish relief supplies and set up new emergency facilities in affected areas, it is necessary to prepare relief supplies in advance to deal with the early stages of an outbreak. In order to predict the number of infected patients in the future, we can build a time-varying diffusion model. Although diffusion dynamics models have been applied to many studies on emergency management, it is unrealistic to predict with perfect accuracy an epidemic. Because the outbreak of an epidemic disease is highly unpredictable, one of the most important goals is to fully consider the uncertainty of the disaster.

For most disasters, providing relief supplies to injured people or infected patients is not enough. Transferring affected people from disaster areas to hospitals ensures they receive better treatment. During an epidemic outbreak, transferring patients is of great significance for maintaining the stability of affected areas and avoiding further outbreaks of infectious diseases. Although there is much research on relief supplies allocation, the interaction between patient transfer and relief distribution has not been well addressed in epidemic outbreaks. In this study, we not only transfer patients to hospitals but also provide these patients with relief supplies.

Because various stakeholders are concerned about emergency management, one of the major challenges of emergency management is to ensure fair emergency relief for all affected people. On the other hand, the unfair distribution of relief supplies may cause panic and psychological stress among the affected people. Furthermore, if the emergency operations only transfer patients without providing relief supplies, it may lead to the deterioration of infected people after transfer.

This paper focuses on emergency management during an epidemic outbreak to ensure that epidemic patients are treated and relief supplies are provided, considering the uncertainties and the control measures. In this study, coordination between patient transfer and relief distribution considers pre-pandemic preparedness measures and post-pandemic relief operations. At the same time, emergency relief plans provide reasonable decision support to decision makers in terms of equity, efficiency, and economics. The contributions of our study are as follows: first, the TMS-PTRD model is proposed to address the locations of TRDCs and the inventory levels of relief supplies in the preparedness state, and to address the locations of THs, patient transfer, and relief distribution for epidemic disasters; second, a multi-objective programming model for patient transfer and relief distribution problems is proposed to balance the efficiency, the fairness, and the economy of the emergency management system; finally, an improved multi-objective algorithm is proposed for large-scale TMS-PTRD problems, and the insights of emergency management are obtained.

The rest of this article is arranged as follows. The Section 2 reviews the literature and analyzes the gaps in the existing research. The Section 3 introduces the problem and constructs the TMS-PTRD model. The Section 4 designs an improved multi-objective algorithm. The epidemic simulations and numerical experiments are carried out in the Section 5. The last section expresses the conclusions and future research.

## 2. Literature Review

In order to reduce the damage caused by disasters, theoretical research and practical applications of emergency management have attracted more attention in recent years. Existing research can generally be divided into two categories: relief distribution and patient transfer.

An important measure of emergency management is relief distribution to the affected areas. Effective distribution of relief supplies (such as medicines, blood supplies, and emergency supplies) is a huge challenge for emergency logistics [5]. Considering population distribution and resource constraints, Duhamel et al. presented a location–allocation model to seek optimal solutions for post-disaster operations [6]. Moreno et al. [7] developed a multi-period, multi-commodity relief location-routing model and proposed a customized heuristics model based on time, stage, and scenario. In this research, vehicle reuse was considered to achieve coordination between different participants in emergency logistics. Because of the transportation system caused by an earthquake, a multi-objective, multi-period relief distribution model considering road repair was developed by Vahdani et al. [8]. Ghasemi et al. [9] considered several types of casualties and failure of emergency centers in an earthquake. The total cost and satisfaction of relief materials were considered in the disaster relief network. Elci et al. [10] considered a relief network problem to determine the capacities and locations of emergency facilities in an uncertain post-disaster environment. In general, emergency management consists of four main phases: mitigation, preparedness, response, and recovery [11]. Grass and Fischer [12] considered that two-stage programming is suitable for some emergency management because some decisions must be made before a disaster, and some decisions can only be made after the disaster. Ahmadi et al. [13] proposed a two-stage humanitarian location-routing model that takes into account random travel time to determine the locations of distribution centers and vehicle path planning. In view of its outstanding performance of two-stage planning in emergency preparedness and response, it has received more and more attention in recent years [14]. Vahdani et al. [15] studied a humanitarian logistics model to provide effective relief distribution supplies after an earthquake. In addition, in order to deal with the uncertain factors caused by the disaster, a robust optimization method was used. Considering primary and secondary disasters, Zhang et al. [16] proposed a multi-stage emergency logistics model to address the relief allocation problem. Mansour et al. [17] considered humanitarian logistics regarding pre-position and post-disaster. They presented a two-stage stochastic programming model for the disaster relief problem under such uncertain disaster scenarios.

Another important operation of disaster relief is to transfer the affected patients, which means transporting infected people from affected areas to shelters or hospitals [18,19]. Considering there is a large number of injured people in public emergencies, emergency managers need to formulate strategies for transferring injured people [20,21]. Flores et al. [22] presented a dynamic transfer model, in which affected people were classified. Goerigket [23] proposed a bus transfer model to move affected people to shelters during a disaster. Lim et al. [24] represented a capacity-constrained network model for the short-notice transfer problem. A greedy heuristic model was improved to optimize maximizing total weighted patient outflow. Shahparvari and Abbasi [25] developed an emergency management model to optimize transfer vehicle scheduling in unstable road scenarios after a disaster. Shahparvari et al. [26] developed a short-notice bushfire transfer model to ensure that the maximum number of critical patients can be safely transferred to shelters. As difficult as the relief distribution, patient transfer needs to deal with multiple uncertainties about the affected areas, time, and the scale of a disaster. Bayram and Yaman [27] studied the patient transfer problem considering the uncertainty caused by the destruction of road networks and shelters. Liang et al. [28] considered that transferring victims to shelters during disasters can keep them safe. Considering the uncertainty in the demands of disasters, they proposed a location and evacuation model based on minimizing total transfer time.

Because patient transfer and relief distribution are considered separately, a single emergency operation has a poor result for emergency management [29]. At present, few articles have considered the importance of combining patient transfer with relief distribution for emergency management. Sabah et al. [30] proposed an integrated affected people transfer and relief distribution model to optimize vehicle scheduling. However, the number of the affected people and the relief demand were considered certain in this research. Generally, accurate information is almost impossible to obtain in a disaster. Setiawan et al. [31] proposed a series of humanitarian logistics models to deal with relief distribution and victim transfer problems after a disaster. This research considered coordination among rescue strategies but did not take into account uncertain parameters.

Emergency management should consider not only costs and benefits but also humanitarian issues [32,33]. In addition, non-profit indicators reflect the humanistic care of emergency management [34]. Fereiduni and Shahanaghi [35] presented an emergency network to make optimal choices about location, allocation, and transfer simultaneously. However, the study did not consider humanitarian issues because only the cost objective was considered. Vahdani et al. [36] proposed an emergency location-routing problem to optimize relief supplies delivery and patient transfer. In this study, a fuzzy credibility theory was used to formulate the fuzzy demand of the patients. However, the study only focuses on the setup cost for ELCs, transportation costs, and the fixed cost for vehicles.

Some researchers have taken into account the uncertainty parameters and humanitarian issues in emergency management. Furthermore, a few studies have not obtained an excellent Pareto solution or designed a suitable algorithm for large-scale cases. Considering the demand of different refugees, Wang et al. developed a location–allocation model to optimize total transfer distance, total cost, and total unmet resources. However, this research converted multiple objectives into one objective using a weighting method and did not obtain the actual Pareto solution [37]. Ghasemi et al. [38] developed a victim transfer and relief distribution model to optimize three objectives: the untreated victims, the satisfaction with emergency supplies, and the total cost. However, the research did not consider transfer time, which is very important in emergency management. Mohammadi et al. [39] proposed a multi-objective location–allocation model to optimize victim transfer and relief distribution after a disaster. This study attempts to decide on setting up relief suppliers, selecting locations of relief distribution centers and emergency centers, and transferring injured people and distributing relief.

Based on this review, the conclusions of relevant studies can be drawn as follows:(1)Many studies have considered victim transfer and relief distribution separately, but few studies have considered the coordination between the two operations.(2)Some studies of emergency management have considered humanitarian issues, but few studies have been able to obtain a real Pareto optimal solution set.(3)No study has considered the characteristics of epidemic diseases and the control measures in a multi-objective victim transfer and relief distribution problem.

Therefore, we propose a two-stage multi-objective stochastic model (TMS-PTRD) considering patient transfer and relief distribution to deal with an epidemic disease. In addition, emergency management is a typical NP-hard problem involving the location of emergency facilities, personnel transfer, material distribution, and vehicle routing. So, it poses a great challenge for multi-objective optimization algorithms. Exact algorithms have been shown to solve small-scale emergency management problems. However, evolutionary algorithms such as NSGA-II [40] and MOPSO [41] have been usually used to solve large-scale multi-objective emergency management problems. When solving problems with more than three objectives, some classical multi-objective optimization algorithms are likely to have poor performance [42]. In order to solve the problem with more objectives, researchers have developed some many-objective algorithms such as MOEA/D [43] and HypE [44]. The principle of the above many-objective algorithms is based on the relationship between multiple objectives rather than the dominant relationship between solutions. A novel concept of decision-maker preference co-evolution has been designed for the many-objective problem (excess of three objectives) [45]. Based on the concept, a preference-inspired co-evolutionary algorithm (PICEA-g) was developed by Wang et al. [46,47]. Previous studies have shown that one heuristic algorithm is usually not suitable for different models [7]. Hence, it is important to design a tailored algorithm for the TMS-PTRD model proposed in this paper.

## 3. Problem Definition and Formulation

The TMS-PTRD model addresses patient transfer and relief distribution during an outbreak. We consider the problem (see Figure 1) as an integrated emergency network, consisting of TRDCs, fever clinics, THs, and DHs. In the preparation phase, the locations of the TRDCs and the inventory levels of relief supplies are determined. After the outbreak, emergency decisions include the locations of the THs, the locations of the DHs, the relief distribution for the fever clinics, and the transfer plan for different types of infected people.

In Figure 1, black arrows represent relief distribution. The blue arrows show the flows of infected people, and the red arrows show the flows of critical patients. The main assumptions are as follows:(1)Each community has a fever clinic, in which the general physicians in the community diagnose illnesses and transfer confirmed cases to the appropriate hospitals;(2)The confirmed cases consisting of the infected people and critical patients can be estimated by the modified SEIR model;(3)We assume that untransferred confirmed cases do not affect the forecast results, because only a small number of confirmed cases are not transferred in time and will be preferentially transferred in the next period;(4)The infected people need to be transferred to the THs and the critical patients need to be transferred to the DHs;(5)The fever clinics provide a set of medical and ancillary supplies (such as an N95 mask and an additional protective suit) for each infected patient and critical patient;(6)Each established TRDC can distribute relief supplies to each fever clinic using a vehicle with a homogeneous capacity.

Note that the planning horizon is a period of time (such as a week) at the beginning of the epidemic, after which the affected areas can get external relief supplies and build more emergency facilities. How measures are implemented goes beyond our research scope.

### 3.1. Time-Varying Epidemic Prediction Model

The relief demand depends on the number of confirmed cases during an epidemic. When making emergency management plans, it is necessary to assess the development of the epidemic in advance. The SEIR models were applicable to epidemic diffusion research [48,49]. Considering epidemic outbreaks are usually short-lived, we assume that the population in the affected areas will not change significantly. In the SEIR model, we assume that the recovered individuals receive lifelong immunity. Hence, ordinary differential equations can be used to describe the epidemic spread as follows.
(1){dStdt=−rβItStdEtdt=rβItSt−ωEtdItdt=ωEt−γItdRtdt=γIt
where N is the total population of the infected areas, *S*(*t*) is the number of susceptible people in an outbreak, *E*(*t*) is the number of exposed people in contact with the infection, *I*(*t*) is the number of infected people with all symptoms, *R*(*t*) is the number of recovered people, r is the exposure ratio, β is the transmission probability, ω is the morbidity of patients in the latent period, and γ is the recovery rate.

For many diseases such as SARS and COVID-19, patients in the latent period can also spread to susceptible people. In practice, transferring confirmed cases to hospitals can control these patients from spreading the disease. Considering the interventions of epidemic diseases, the susceptible people in isolation (SI), the exposed people in isolation (EI), and the critical patients (H) in urgent need of hospitalization are added to the modified model. Based on the above point of view, we use a modified SEIR model to evaluate the epidemic [50].
(2){dS(t)dt=−[rβ+rq(1−β)]S(t)[I(t)+θE(t)]+λSq(t)dE(t)dt=rβ(1−q)S(t)[I(t)+θE(t)]−ωE(t)dI(t)dt=ωE(t)−(δI+κ+γI)I(t)dSI(t)dt=rq(1−β)S(t)[I(t)+θE(t)]−λSI(t)dEI(t)dt=rβqS(t)[I(t)+θE(t)]−δqEI(t)dH(t)dt=δII(t)+δqE(t) −(κ+γH)H(t)dR(t)dt=γII(t)+γHH(t)
where q is the isolation ratio, θ is the transmission coefficient in the latent period, λ is the rate of release from quarantine, κ is the death rate, $\delta_{q}$ is the morbidity of quarantined patients in the latent period, $\delta_{I}$ is the probability that infected people will be isolated for treatment, $\gamma_{H}$ is the recovery rate of hospitalized patients, and $\gamma_{I}$ is the recovery rate of infected people.

Given the initial values and related parameters, the modified SEIR model can be used to assess the time series data of patients. We define the relief supplies required by each patient in a day as *θ*. The infected people need to be transferred to the THs for isolation and treatment, and the critical patients need to be transferred to the DHs for specialty treatment [50]. Before reaching the hospitals, relief supplies must be provided to the infected people and critical patients to ensure that the disease is not spread during transfer. So, the demand for relief supplies is formulated as:(3)$$d=\theta \left[I\left(t\right)+H\left(t\right)\right]$$

### 3.2. Notation

Sets and Index:
S:  Set of scenarios, s∈S;T:Set of planning periods, t∈T;I:Set of TRDCs, i∈I;C:Set of fever clinics, c∈C;J:Set of THs, j∈J;H:Set of DHs, h∈H.

Model Parameters:
$P_{s}$:Probability of scenario *s*;$t_{nm}^{p}$:Transportation time from the node n to the node *m*;$Fc_{i}$:Fixed cost of the TRDC *i*;$Ft_{j}$:Fixed cost of the TH *j*;$Fd_{h}$:Fixed cost of the DH *h*;$c_{1}$:Holding cost of a relief supply;$c_{2}$:Penalty cost of a unit of relief supply;$C_{cj}^{M}$:Transfer cost for the infected people from the fever clinic *c* to the TH *j*;$C_{ch}^{S}$:Transfer cost for a critical patient from the fever clinic *c* to the DT *h*;$CV_{ic}^{R}$:Transport fees for one vehicle from the TRDC *i* to the fever clinic *c*;$CR_{i}$:Maximum capacity of the TRDC *i*;Cav:Capacity of a vehicle to carry the relief supplies;$Cm_{j}$:Maximum capacity of the TH *j*;$Cs_{h}$:Maximum capacity of the DH *h*;$\tilde{I}p_{c}^{ts}$:Number of infected people at the fever clinic *c* in scenario *s*;$\tilde{C}p_{c}^{ts}$:Number of critical patients at the fever clinic *c* in scenario *s*;$\phi^{S}$:Priority of satisfying the critical patient; α:Confidence levels of the chance-constrained model;$M_{big}$:A large positive number.

Decision Variables:
$Y_{i}$:1, if the TRDC *i* is established, 0 else;$Z_{j}^{s}$:1, if the TH *j* is established under scenario *s*, 0 else;$K_{h}^{s}$:1, if the designated hospital h be established under scenario *s*, 0 else;$VN_{i}$:Amount of inventory at the TRDC *i*;$X_{ic}^{ts}$:Number of relief supplies transported from the TRDC *i* to the fever clinic *c* in scenario *s*.

### 3.3. Model Formulation

Equation (4) minimizes the number of patients not transferred at different periods. Equation (5) minimizes the transfer time. Equation (6) minimizes the fixed cost of the TRDCs and pre-stored relief supplies in the first phase. In the second phase, the objective function minimizes the fixed THs, the fixed DHs, the cost of distributing relief supplies, and the cost of transferring patients. Equation (7) minimizes the dissatisfaction with relief service. Equations (8)–(10) indicate only one emergency facility can be established at each point. Equation (11) describes that only established TRDCs can be applied. Equation (12) describes the amount of unused relief supply. Equation (13) describes the carrying capacity of the transportation vehicles. Equation (14) describes the capacity of established THs. Equation (15) describes the capacity of the designated hospitals. Equation (16) defines the unserved confirmed patients in the fever clinics. Equation (17) defines a satisfaction function for the fever clinics. Equations (18) and (19) describe the binary variables and the non-negative variables.
(4)$$\min f_{1}=\sum_{s\in S}\sum_{t\in T}\sum_{c\in C}P_{s}\cdot \left[\left(\sum_{\tau =1}^{t}\tilde{I}p_{c}^{\tau s}-\sum_{\tau =1}^{t}\sum_{j\in J}PM_{cj}^{\tau s}\right)+\phi^{S}\left(\sum_{\tau =1}^{t}\;\tilde{C}p_{c}^{\tau s}-\sum_{\tau =1}^{t}\sum_{h\in H}PS_{ch}^{\tau s}\right)\right]\;$$
(5)$$\min f_{2}=\sum_{s\in S}P_{s}\cdot \left(\sum_{t\in T}\sum_{c\in C}\sum_{j\in J}t_{cj}PM_{cj}^{ts}+\phi^{S}\sum_{t\in T}\sum_{c\in C}\sum_{h\in H}t_{ch}PS_{ch}^{ts}\right)$$
(6)$$\begin{array}{l} \min f_{3}=\sum_{i\in I}Fc_{i}\cdot Y_{i}+\sum_{i\in I}c_{1}\cdot VN_{i}+\sum_{s\in S}P_{s}\cdot \left(\sum_{j\in J}Ft_{j}\cdot Z_{j}^{s}+\sum_{h\in H}Fd_{h}\cdot K_{h}^{s}+\sum_{t\in T}\sum_{i\in I}\sum_{c\in C}CV_{ic}^{R}VR_{ic}^{ts}+\right.\sum_{t\in T}\sum_{c\in C}\sum_{j\in J}C_{cj}^{M}PM_{cj}^{ts}+\\ \left.\sum_{t\in T}\sum_{c\in C}\sum_{h\in H}C_{ch}^{S}PS_{ch}^{ts}+\sum_{i\in I}c_{2}\cdot UVN_{i}^{s}\right) \end{array}$$
(7)$$\min f_{4}=\sum_{s\in S}\sum_{t\in T}\sum_{c\in C}P_{s}\cdot \varphi \left(U_{c}^{ts}\right)$$
(8)$$Y_{i}\leq 1,\forall i\in I$$
(9)$$Z_{j}^{s}\leq 1,\forall j\in J$$
(10)$$K_{h}^{s}\leq 1,\forall h\in H$$
(11)$$VN_{i}\leq CR_{i}\cdot Y_{i},\forall i\in I\;$$
(12)$$VN_{i}-\sum_{t\in T}\sum_{c\in C}X_{ic}^{ts}\leq UVN_{i}^{s},\forall i\in I,s\in S$$
(13)$$X_{ic}^{ts}\leq cav\cdot VR_{ic}^{ts},\forall c\in C,i\in I,t\in T,s\in S$$
(14)$$\sum_{t\in T}\sum_{c\in C}PM_{cj}^{ts}\leq Cm_{j}Z_{j}^{s},\forall j\in J,s\in S$$
(15)$$\sum_{t\in T}\sum_{c\in C}PS_{ch}^{ts}\leq Cs_{h}K_{h}^{s},\forall h\in H,s\in S$$
(16)$$U_{c}^{ts}\leq 1-\frac{\sum_{i\in I}X_{ic}^{ts}}{\tilde{I}p_{c}^{ts}+\tilde{C}p_{c}^{ts}},\forall c\in C,t\in T,s\in S$$
(17)$$\varphi \left(U_{c}^{ts}\right)=e^{\frac{-U_{c}^{ts}}{1-U_{c}^{ts}}},\forall c\in C,t\in T,s\in S$$
(18)$$Y_{i},Z_{j}^{s},K_{h}^{s}\in \left\{0,1\right\},\forall i\in I,j\in J,h\in H,s\in S$$
(19)$$VN_{i},X_{ic}^{ts},VR_{ic}^{ts},UVN_{i}^{s},PM_{cj}^{ts},PS_{ch}^{ts}\geq 0\&Integer,\forall c\in C,i\in I,j\in J,h\in H,t\in T,s\in S$$

### 3.4. A Chance-Constrained Model for Patient Transfer and Relief Distribution

Considering the results predicted by the modified SEIR model will partially deviate from the actual data, this may have stochastic characteristics in the two-stage patient transfer and relief distribution problem. The stochastic model is provided with a predetermined reliability level (*α*) by using chance constraints. So, we used the chance-constraint model to define reliable sets as follows [51].
(20)$$\underset{x\in X}{\min}f_{k}\left(x\right)=\sum_{j=1}^{n}{\tilde{c}}_{kj}x_{j},k=1,2,...,K$$
(21)$$\mathrm{Pr}\left\{\sum_{j=1}^{N}a_{ij}x_{j}\geq {\tilde{b}}_{i}\right\}\geq \alpha_{i},i=1,2,...,M$$
(22)x∈X
where X is the deterministic feasible region, *f(x)* is the value of the minimized function, and α is the confidence level of the chance-constrained optimization. According to the inverse function theorem, the chance constraint is rewritten are as follows:(23)$$\underset{x\in X}{\min}f_{k}\left(x\right)=E\left(\sum_{j=1}^{N}{\tilde{c}}_{kj}x_{j}\geq {\tilde{b}}_{i}\right),k=1,2,...,K;i=1,2,...,M$$
(24)$$\mathrm{Pr}\left\{\sum_{i=1}^{I}{\tilde{a}}_{ij}x_{i}\geq {\tilde{b}}_{i}\right\}\geq \alpha_{i},i=1,2,...,M$$
(25)$$x=\left(x_{1},x_{2},...,x_{n}\right)$$
(26)x∈X

Based on the above theory, the summary can be rewritten as follows.
(27)$$E\left(\sum_{j=1}^{n}c_{kj}^{*}x_{j}-f_{k}^{-}\right)-\varphi^{-1}\left(\alpha_{k}\right)\sqrt{Var}\left(\sum_{j=1}^{n}c_{kj}^{*}x_{j}-f_{k}^{-}\right)\geq 0,k=1,2,...,K$$
(28)$$E\left(\sum_{j=1}^{n}c_{kj}^{*}x_{j}-f_{k}^{+}\right)+\varphi^{-1}\left(\alpha_{k}\right)\sqrt{Var}\left(\sum_{j=1}^{n}c_{kj}^{*}x_{j}-f_{k}^{+}\right)\leq 0,k=1,2,...,K$$
where $f_{k}^{-}=\min c_{kj}^{*}x_{j}$ and $f_{k}^{+}=\max c_{kj}^{*}x_{j}$.
(29)$$E\left(\sum_{j=1}^{n}{\tilde{a}}_{ij}x_{j}-{\tilde{b}}_{i}\right)-\varphi^{-1}\left(1-\alpha_{i}\right)\sqrt{Var}\left(\sum_{j=1}^{n}{\tilde{a}}_{ij}x_{j}-{\tilde{b}}_{i}\right)\geq 0,i=1,2,...,M$$

Making use of the auxiliary variables $W_{stc}^{1}$ and $W_{stc}^{2}$, we rewrite the first objective function as a deterministic model at the confidence level α as follows:(30)$$\min f_{1}=\sum_{s\in S}\sum_{t\in T}\sum_{c\in C}P_{s}\cdot \left(W_{stc}^{1}+\phi^{S}W_{stc}^{2}\right)$$
(31)$$E\left(\sum_{\tau =1}^{t}\bar{I}p_{c}^{ts}\right)+\varphi^{-1}\left(1-\alpha_{j}\right)\sqrt{Var}\left(\sum_{\tau =1}^{t}\tilde{I}p_{c}^{ts}\right)-\sum_{\tau =1}^{t}\sum_{j\in J}PM_{cj}^{\tau s}\leq W_{stc}^{1}$$
(32)$$E\left(\sum_{\tau =1}^{t}\bar{C}p_{c}^{\tau s}\right)+\varphi^{-1}\left(1-\alpha_{j}\right)\sqrt{Var}\left(\sum_{\tau =1}^{t}\tilde{C}p_{c}^{\tau s}\right)-\sum_{\tau =1}^{t}\sum_{h\in H}PS_{ch}^{\tau s}\leq W_{stc}^{2}$$
(33)$$W_{stc}^{1},\;W_{stc}^{2}\geq 0$$

Similarly, Equation (16) is modified as follows:(34)$$U_{c}^{ts}\leq 1-\frac{\sum_{i\in I}XS_{ic}^{ts}}{\left(\bar{I}p_{c}^{ts}+\bar{C}p_{c}^{ts}\right)+\varphi^{-1}\left(1-\alpha_{j}\right)\sqrt{Var}\left(\tilde{I}p_{c}^{ts}+\tilde{C}p_{c}^{ts}\right)},\forall c\in C,t\in T,s\in S$$

Finally, the chance-constrained model for the TMS-PTRD problem is represented as follows.

Min. (30), (5), (6), and (7)

S.t. (8)~(15), (17)~(19), (31)~(34).

## 4. Solution Algorithm

### 4.1. ε-Constraint Algorithm

In the ε-constraint algorithm, the highest priority objective is selected based on the preferences of the decision maker, and other objectives are converted to constraints. The Pareto front is obtained by changing the upper and lower boundaries of these constraints [52]. The ε-constraint algorithm is as follows:(35)$$\min f_{1}\left(X\right)$$

S.t.
(36)$$f_{2}\left(X\right)\leq \varepsilon_{2}$$
(37)$$f_{m}\left(X\right)\leq \varepsilon_{m}$$
where *X* is a feasible set. We use the epsilon parameters to show the range of each objective function. The upper and lower bounds of the objective function are shown in Equations (38) and (39).
(38)$${\bar{\varepsilon }}_{i}=\min\left\{f_{i}\left(x\right)\left|x\in X\right.\right\}$$
(39)$${\underline{\varepsilon }}_{i}=\max\left\{f_{i}\left(x_{1}^{*}\right),f_{i}\left(x_{2}^{*}\right),...f_{i}\left(x_{m}^{*}\right)\right\}$$

We use Equation (40) to calculate the ε parameters.
(40)$$\varepsilon_{ij_{i}}={\underline{\varepsilon }}_{i}-\left(\frac{{\underline{\varepsilon }}_{i}-{\bar{\varepsilon }}_{i}}{q_{i}}\right)\cdot j_{i},j_{i}=0,1,...,q_{i}$$
where $q_{i}$ represents the number of intervals between the upper and lower bounds. In this algorithm, a payoff constrain table of the ε-constraint algorithm is as follows:

Step 1:Calculate a payoff table $$\left(\begin{array}{c} \begin{array}{cccc} f_{2}\left(x_{1}^{*}\right) & f_{3}\left(x_{1}^{*}\right) & ... & f_{m}\left(x_{1}^{*}\right) \end{array}\\ ...\\ \begin{array}{cccc} f_{2}\left(x_{2}^{*}\right) & f_{3}\left(x_{3}^{*}\right) & ... & f_{m}\left(x_{m}^{*}\right) \end{array} \end{array}\right)$$
$$x_{i}^{*}=\arg_{x\in X}\min f_{i}\left(x\right),i=1,2,...,m$$

Step 2:Set ${\bar{\varepsilon }}_{i}=f_{i}\left(x_{i}^{*}\right),{\underline{\varepsilon }}_{i}=f_{i}\left(x_{1}^{*}\right),i=1,2,...,m$ a

Step 3:

$P=\left\{x_{1}^{*},x_{2}^{*},...,x_{m}^{*}\right\},F=\left\{f_{1}\left(x_{1}^{*}\right),f_{2}\left(x_{2}^{*}\right),...,f_{m}\left(x_{m}^{*}\right)\right\}$



Step 4:Solve $x^{*}=opt\left(f_{1},\varepsilon_{2j_{2}},\varepsilon_{3j_{3}},...,\varepsilon_{mj_{m}}\right)$
$$f_{2}\left(X\right)\leq \varepsilon_{2j_{2}}\;\mathrm{for}\;j_{2}=0:1:q_{2}\;f_{3}\left(X\right)\leq \varepsilon_{3j_{3}}\;\mathrm{for}\;j_{3}=0:1:q_{3}\;\ldots \;f_{m}\left(X\right)\leq \varepsilon_{mj_{m}}\;\mathrm{for}\;j_{m}=0:1:q_{m}$$

Step 5:

$P=P\cup \left\{x^{*}\right\},F=F\cup \left\{f_{1}\left(x^{*}\right),f_{2}\left(x^{*}\right),...,f_{m}\left(x^{*}\right)\right\}$



Step 6:Return *P* and *F*

Given the priority of the objective functions, the ε-constraint algorithm can obtain the Pareto front of multi-objective problems.

### 4.2. An Improved PICEA-g Algorithm

Evolutionary algorithms have been applied to solve multi-objective problems. Numerous studies have shown that utilizing neighborhood information can make significant progress in algorithm research [53,54]. Considering it is very difficult to define the neighborhood of the TMS-PTRD model using Euclidean distance, a novel similarity distance is designed to construct the K-nearest neighborhood. The PICEA-g-AKNN algorithm is proposed by integrating the presented K-nearest neighborhood into the framework of the preference-inspired co-evolutionary algorithm (PICEA-g). At the same time, we use an evaluation method according to the fitness of the solutions and assign a tailored evolutionary strategy for each type of chromosome to improve the efficiency of the algorithm.

#### 4.2.1. An Adaptive K-Nearest Neighborhood Method Based on a Novel Similarity Distance

We introduce an intuitive definition for similarity distance to determine the neighborhood for each chromosome. The two types of similarities between chromosomes are defined as follows:The similarity of the locations of emergency facilities

This type of similarity between X and $X^{\prime }$ are measured by the location of the emergency facilities. Considering two factors (locations and size), the similarity of the emergency facilities is formulated as:(41)$$SD_{1}(X,X^{\prime })=\frac{1}{\left|I\right|\left|L\right|}\sum_{i\in I}\sum_{l\in L}\varphi_{1}\left(x_{il},x_{il}^{\prime }\right)$$
(42)$$\varphi_{1}\left(x_{i},x_{i}^{\prime }\right)=\left\{\begin{array}{c} 1,\;if\;x_{il}=x_{il}^{\prime }\\ 0,\;otherwise\; \end{array}\right.$$
where $x_{il}$ is one if emergency facility I is established; otherwise, $x_{il}$ is zero. The example in Figure 2 illustrates the encoding process for emergency facility locations.

In Figure 2, Chromosome (a) chooses to establish the second TRDC at Size 1 and the fourth TRDC at Size 3. Chromosome (b) chooses to establish the second TRDC at Size 1, the third TRDC at Size 2, and the fourth TRDC at Size 2.
2The similarity between distribution planning and transfer planning

The similarity of the patient flows between solution X and solution $X^{\prime }$ can be measured by the number of patients transferred. Similarly, the similarity of the relief flows is measured by the transportation quantity of relief supplies.
(43)$$SD_{2}(X,X^{\prime })=\sum_{i\in N}\sum_{j\in M}\frac{1}{\left|N\right|\left|M\right|}\varphi_{2}\left(\alpha_{stij},\alpha_{stij}^{\prime }\right)$$

At the decision period t, $\alpha_{stij}$, and $\alpha_{stij}^{\prime }$ are the number of relief supplies transported from the TRDC i to the clinic j in scenario s.
(44)$$\varphi_{2}\left(\alpha_{stij},\alpha_{stij}^{\prime }\right)=\left\{\begin{array}{c} \alpha_{stij}/\alpha_{stij}^{\prime },\;if\;\alpha_{stij}^{\prime }>\alpha_{stij}\;\\ \alpha_{stij}^{\prime }/\alpha_{stij},\;if\;\alpha_{stij}>\alpha_{stij}^{\prime }\;\\ \;1,\;if\;\alpha_{stij}=\alpha_{stij}^{\prime }\; \end{array}\right.$$

The example in Figure 3 illustrates the encoding process of the patient flows and the relief flows.

From Figure 3, 100 relief supplies are transported from the second TRDC to Clinic 2, and 300 relief supplies are transported from the second TRDC to Clinic 4 in Chromosome (a). Similarly, 300 relief supplies are transported to Clinic 2 and Clinic 3.

Considering the two-stage stochastic model, we construct the similarity distance between two chromosomes as follows:(45)$$SD(X,X^{\prime })=SD_{1}(X,X^{\prime })+\frac{1}{\left|S\right|\left|T\right|}\sum_{s\in S}\sum_{t\in T}SD_{2}(X,X^{\prime })$$

The K-nearest neighborhood has been proven useful in the field of computer science and engineering [55,56]. Generally speaking, a smaller neighborhood size can enhance local search ability, while a larger neighborhood size can improve global search ability [57]. So, an adaptive K-nearest neighborhood method is used to dynamically adjust the neighborhood size to balance exploitation ability and exploration ability. By evaluating the evolutionary state of the solutions, the adaptive K-nearest method decreases the neighborhood radius *k* to enhance the exploitation ability when the solution is in a continuous evolution state and increases the neighborhood radius k to improve the exploration ability when the solution is in a state of stagnant evolution. The adaptive K-nearest construction method is as Algorithm 1.

**Algorithm 1:** Framework of the K-nearest neighborhood method**Input:** solution S, neighborhood size *K*_1_ and *K*_2_
**1.** set the current neighborhood size *K**_c_* and *K*_1_;
**2.****for**gen=1:genMax;
**3.**          Find non-dominated solutions *_NS_* in *_S_*;
**4.**            Generate offspring solutions *_OS_* based on neighborhood size *K**_c_*;
**5.**            **if**mod(gen,5)==0**6.**                  Find non-dominated solutions *_NOS_* in *_OS_*;
**7.**                  **if**$x^{\prime }≺x,\;x\in S,\;x^{\prime }\in OS$**8.**                      $K_{c}=K_{c}-1$;
**9.**                  **else****10.**                    $K_{c}=K_{c}+1$;
**11.**                  **end if****12.**              **end if****13.**              **if**$K_{c}>K_{1}$**14.**                $K_{c}=K_{1}$;
**15.**                **else****16.**                $K_{c}=K_{2}$**17.**              **end if**
**18.**
**end for**


#### 4.2.2. The PICEA-g-AKNN Algorithm

In the presented algorithm, all solutions are divided into three evolutionary states by an assessment method. In addition, each state is assigned a customized evolutionary strategy. The assessment method and crossover operation for individuals are as follows.

1We define a dominant individual if the solution dominates all of the neighborhoods. It is probable that the dominant individual is close to the Pareto front. Therefore, an SBX local search strategy is used to modify this individual. The SBX local search strategy not only enables the dominant individual to move close to the Pareto fronts but also does not cause a large disturbance for the outstanding individual. Let Pc is the probability of executing SBX and the SBX is described as Equation (46).
(46)$$cx_{il}=\{\begin{array}{c} 0.5\cdot \left[\left(1+\beta \right)x_{il}+\left(1-\beta \right)x_{jl}\right],\;if\;rc\leq Pc\\ 0.5\cdot \left[\left(1+\beta \right)x_{il}-\left(1-\beta \right)x_{jl}\right],\;otherwise \end{array}$$
where rc is a randomly generated number in the range [0, 1]. Note that if $cx_{il}\leq 0$, we assign that $cx_{il}$ is equal to zero. If $x_{il}=x_{jl}$, then assign $cx_{il}$ to $x_{il}$, where $cx_{il}$ is the offspring $x_{i}$ in the l dimension, $x_{j}$ is a parent chromosome selected at random from the neighborhood, and β is a random number generated as follows according to [58].
(47)β={ (2⋅rand)11+η, if rand≤0.5[1/(2−2⋅rand)]11+η, otherwise
where η is a parameter that represents the degree of learning from the parent individual.


2A solution is defined as an exploring individual if it is not dominated by any neighborhood and there are other non-dominant individuals in the neighborhood. The exploring individual is likely to obtain useful information from the non-dominant individual in the neighborhood. We use a classic and robust DE operator named “DE/rand/1” to generate offspring. Because more parents can be referenced in the DE, children individuals can exchange excellent information with individuals in the neighborhood [59]:(48)$$cx_{il}=x_{il}+F\cdot \left(x_{r1,l}-x_{r2,l}\right)$$
where $cx_{i}$ is the children individual of $x_{i}$, and $x_{r1}$ and $x_{r2}$ are randomly selected from the non-dominant individuals in the neighborhood. When there is only one non-dominant solution of $x_{i}$ in the neighborhood, the other parent individual is selected randomly from the other individuals in the neighborhood. F∈(0,2) is the mutation factor. We assign $cx_{il}$ is equal to zero if $cx_{il}\leq 0$, and assign $cx_{il}$ to $x_{il}$ if $x_{il}=x_{jl}$.


3A solution is defined as a learning individual if it is dominated by individuals in the neighborhood. For the learning individual, it is necessary to try to learn from the outstanding individuals in the neighborhood. A directional search DE named “DE/current-to-dominance” has good global search ability and is used to generate offspring [60].
(49)$$cx_{il}=x_{il}+K\cdot \left(x_{dl}-x_{il}\right)+F\cdot \left(x_{r1,l}-x_{r2,l}\right)$$
where $x_{d}$ is an individual that dominates the current individual in the neighborhood, K∈(0,1) is another mutation factor, and $x_{r1}$ and $x_{r2}$ are randomly selected from the individuals that dominate $x_{i}$. If only one solution dominates $x_{i}$, the other parent individual can choose from the non-dominated solution of $x_{i}$. We assign $cx_{il}$ when it is equal to zero if $cx_{il}\leq 0$, then assign $cx_{il}$ to $x_{il}$ if $x_{il}=x_{jl}$.


In the PICEA-g-AKNN algorithm, the fitness value of the candidate solution is positively correlated with the number of goal vectors dominated by this candidate solution, but the fitness value of the target vector is inversely proportional to the number of candidate solutions dominating this target vector. The PICEA-g-AKNN algorithm is shown in Figure 4.

## 5. Computational Examinations

For validating the TMS-PTRD model, a case study is presented based on coronavirus disease 2019 (COVID-19) in Wuhan, Hubei, China. Moreover, we design a series of TMS-PTRD models to verify the PICEA-g-AKNN algorithm.

### 5.1. Simulation of Forecasting Phase

This part uses a modified SEIR epidemic dynamics model to evaluate COVID-19 in Wuhan. Wuhan, which covers an area of 8483 km^2^ and has more than 10 million permanent residents, was one of the most seriously affected areas during the COVID-19 outbreak. Since 23 January 2020, the urban area of Wuhan has been effectively blocked. According to the Wuhan Novel Pneumonia Prevention and Control Headquarters, some temporary hospitals were established to receive infected people with mild symptoms, and some designated hospitals were selected to treat crucial patients. In Wuhan city, the population size within the lockdown area has not changed significantly. Because of the intervention measures, people were asked to stay at home to minimize contact with pedestrians.

The actual data were selected from the daily data of the COVID-19 epidemic in Wuhan from 21 January to 21 April 2020 (wjw.hubei.gov.cn/, accessed on 25 April 2020). The initial values were chosen as: N = 1,2320,000, E(0) = 149, I(0) = 440, R(0) = 28, H(0) = 102, SI(0) = 2197, EI(0) = 1394, S(0) = N−E(0)−I(0)−R(0)−H(0)−SI(0)−I(0). In line with Chao et al. [50], the parameters are shown in Table 1.

Note that because of changes in statistical methods, official figures of the cases have risen sharply since 12 February, and the model parameters have been modified appropriately. Figure 5 shows that the number of confirmed patients predicted by the SEIR model reaches a maximum at around 19 February. Considering the results show that the correlation coefficient between the predicted number and the actual number is more than 90%, there is no statistically significant difference between the actual number and the predicted number. So, this forecasting accuracy is enough for emergency management decision making.

### 5.2. Simulation of the Second Stage

Suppose that the lockdown city has 10 affected areas, and each affected area has a fever clinic where the general physicians diagnose illnesses and transfer confirmed cases to the appropriate hospitals. There are three TRDCs, three THs, and three DHs. We set the planning horizon to one week, taking into account the availability of external relief supplies and the additional emergency facilities in the affected areas for some time after the epidemic outbreak. Because it is difficult to obtain the start date of the epidemic, a certain time in the past is selected as the start date of the planning period. In a scenario, the parameters of the affected areas are the same as Table 1, and the initial values are shown in Table 2.

The parameters of the above scenarios are based on the implementation of strict control measures during the COVID-19 outbreak in Wuhan. So, hospital patients do not cause infection, and the exposure ratio of the patient is zero. Because the exposure ratio is likely to increase under conditions of inadequate intervention measures, we attempt to assess scenarios with inadequate measures. In addition, there is a relationship between the value of q and the number of infected cases. To analyze the quarantine during the COVID-19 outbreak, different isolation ratio scenarios are considered in the TMS-PTRD model. We consider the case study consisting of four scenarios: S_1_ (r1=r, q1=q), S_2_ (r1=2r, q1=q), S_3_ (r1=r, q1=0.8q), and S_4_ (r1=2r, q1=0.8q). Each scenario is assumed to occur with equal probability ($p_{s}=\frac{1}{\left|S\right|}$). Within the next week, the number of infected people and critical patients in Disaster Area 1 is calculated by the modified SEIR model, as shown in Table 3.

Assume that the transportation cost ($c_{nm}$) and the delivery time ($t_{nm}$) from point *n* to point *m* are linear. The information is given in Table 4.

A standard carton is used as a transportation unit, including a set of medical and ancillary supplies. The loading capacity of the transportation vehicle is 50. Table 5 and Table 6 describe the parameters of the hospitals and the TRDCs, respectively.

In this section, experimental results are provided. The initial values of the affected areas were set as in Table 2, and the predicted values are shown in Table 3. In this case, the key parameters are shown in Table 4, Table 5 and Table 6. The model was coded using CPLEX and MATLAB software version R2019b, and the numerical experiments were run on a PC with 8 GB of RAM. In this study, we set the priority of critical patients to five. Considering the correlation coefficient between the predicted number and the actual number is more than 90%, the variation domain/perturbation level is set to 10% and the budget parameter of the confidence level is set to 95%.

Figure 6 and Figure 7 show the results of solving a small-scale multi-objective problem by the two methods of the ε-constraint algorithm and the PICEA-g-AKNN algorithm, respectively. In the test problem, the first objective function is on a scale of 10^2^, and the second function is on a scale of 10^3^. Similarly, the third objective function is on a scale of 10^5^, and the fourth function is on a scale of 10^−1^.

From Figure 6 and Figure 7, we calculated that the mean error value of the two algorithms in the first objective was 0.67%. Similarly, the mean error values of the second, third, and fourth objectives were 0.55%, 1.02%, and 0.78%, respectively. So, the PICEA-g-AKNN algorithm is credible in solving TMS-PTRD problems.

To analyze the uncertain parameters, the TMS-PTRD problems were tested under different uncertain conditions, and the output of the TMS-PTRD implementation is shown. In Figure 8, the main objective function increases when the uncertain environment gets worse. The results also show that an accurate assessment of the uncertainty can positively impact the results, where a is the variation domain/perturbation level, and α is the budget parameter of the confidence level.

Table 7 shows the relief flows sent from the established TRDCs to the fever clinics for all periods (a=0.1, α=0.95). For example, TRDCs 1 and 2 are established, and relief supplies are distributed to 10 fever clinics through these two established TRDCs.

Table 8 and Table 9 show the suitable locations for the established THs and DHs in each scenario. For example, all of the THs (three temporary hospitals) are established under Scenario 4, and we just need to establish two THs under Scenarios 1, 2, and 3. In addition, Table 8 and Table 9 determine the number of infected people and critical patients transferred from each fever clinic to hospitals for all periods (a=0.1, α=0.95).

### 5.3. Computational Performance Analysis

We consider the 20, 30, and 50 affected areas, that is, the number of fever clinics that should be serviced. For each affected area, the parameters of the modified SEIR were randomly selected from an interval. Eighteen instances were randomly generated to test the performance of the PICEA-g-AKNN algorithm. In Table 10, for example, T-4-20-4-4-7-4 means that the case with three periods of two uncertain scenarios has 4 TRDCs, 20 fever clinics, 4 THs, and 4 DHs. Initial values of the disaster areas were randomly generated in the intervals shown in Table 11.

Without loss of generality, we set the variation perturbation level at 10% and the budget parameter of the confidence levels at 95%. Experimental results were obtained by a personal computer with 2.50 GHz and 8 GB of RAM. Each test instance was run 10 times to obtain the approximate solutions. The experimental results of the PICEA-G-AKNN algorithm and PICEA-g, MOEA/D, and NSGA-II were compared. The parameters of the algorithms are as Table 12.

Set coverage (C-metric) is used to compare the efficiency of the multi-objective algorithms [61]. In a minimization problem, the C-metric can be represented as follows.
(50)$$C\left(A,B\right)=\frac{\left\{y\in B\left|\exists x\in A:x≺y\right.\right\}}{\left|B\right|}$$
where A is Pareto solutions obtained by one multi-objective algorithm, and B is Pareto solutions obtained by another multi-objective algorithm. The C-metric mean values of the numerical experiments are shown in Table 13.

From the results, the PICEA-g-AKNN algorithm outperforms the other three multi-objective algorithms in solving the TMS-PTRD problems. In particular, the PICEA-g-AKNN algorithm performs well in solving the TMS-PTRD problems with a large number of scenarios and periods. Considering the characteristics of the multi-scenario and multi-period problems, the novel neighborhood method proposed in Section 4.2.1 can reasonably construct a neighborhood of the TMS-PTRD problems. In addition, in the PICEA-g-AKNN algorithm, an appropriate method is used to evaluate the multi-objective solutions, and different evolutionary strategies are assigned to different types of chromosomes, which can effectively improve the efficiency of the multi-objective algorithm. We believe the proposed algorithm has advantages in solving multi-objective problems with multi-scenario and multi-period problems.

To analyze the epidemic interventions in the lockdown area of COVID-19, we considered different levels of epidemic interventions in the lockdown area. When strict intervention is used, the isolation ratio q increases but the exposure rate r decreases. Conversely, lax intervention leads to a decrease in the isolation ratio and an increase in the exposure rate. We tested T-8-50-8-8-14-12 by the PICEA-g-AKNN algorithm and showed the results as follows.

From Figure 9, Figure 10, Figure 11 and Figure 12, we can see that with the increase in the exposure rate r, the first three objectives increase significantly. However, the increase in the isolation ratio q has no obvious effect on the results. We can conclude that reducing the average number of daily contacts and increasing the proportion of quarantined can reduce the expectation of the objectives. The average number of contacts has a more important impact on the development of the epidemic than the proportion quarantined. So, the first three objectives are better when strict intervention has been used. As can be seen from the fourth objective, the presented research can provide reliable equitable relief distribution under different epidemic interventions.

In this study, the TMS-PTRD model and the PICEA-g-AKNN algorithm have the advantages in tackling public health emergencies: (1) A reservation mechanism of relief supplies in densely populated urban areas is very important for epidemic intervention; (2) A relatively accurate method of epidemic prediction can support the decision of emergency management, for instance, setting up a reasonable number of temporary hospitals in different scenarios; (3) The TMS-PTRD model can be used to coordinate patient transfer and relief distribution problems, and multi-objective results can support decision makers to make decisions in their preferences and actual situations; (4) A multi-objective algorithm developed according to the TMS-PTRD model structure is helpful to obtain better Pareto solutions; (5) Reducing person-to-person contact has a positive significance for emergency management in the epidemic.

## 6. Conclusions 

In response to public health emergencies, scientific and reasonable emergency relief can reduce epidemic disaster losses. Because of the suddenness and unpredictability of an epidemic outbreak, the TMS-PTRD model has advantages in pre-disaster preparation and post-disaster rescue. Considering that humanitarian factors cannot be ignored in emergency relief, timeliness and fairness are even more important than economy in emergency rescue. In order to ensure that the study can be more realistic, uncertainty factors and epidemic control measures were considered. This paper investigated the patient transfer and relief distribution problem in lockdown areas of COVID-19, which was depicted by a two-stage multi-objective stochastic model. The proposed model considers the following objectives: the total number of untreated infected patients at different periods, the total transfer time, the overall cost, and the equity distribution of relief supplies. To better cope with epidemic disasters, this paper considered setting up emergency facilities and storing a certain number of emergency supplies before an epidemic outbreak occurs. After an outbreak, the transfer plans of patients and the relief distribution are determined. Considering stochastic parameters, this study used chance-constraint programming to define the condition of a reliable set. To efficiently solve the large-scale TMS-PTRD problem, we proposed the PICEA-g-AKNN algorithm based on a novel similarity distance and tailored evolutionary strategies.

A real-world case study of Hunan of China and 18 test instances were generated to evaluate the TMS-PTRD model and the proposed algorithm. Numerical experiments show that the proposed method can well meet the needs of infected patients in an epidemic outbreak. In other words, hospital beds and relief supplies can be provided to infected patients in a timely manner. At the same time, numerical experiments showed that exposure reduction is more effective than other control measures. In this paper, the improved multi-objective algorithm can effectively solve large-scale TMS-PTRD problems. Experimental results show that the proposed PICEA-g-AKNN algorithm outperforms the PICEA-g, MOEA/D, and NSGA-II algorithms in solving the TMS-PTRD model. Moreover, the proposed TMS-PTRD model and PICEA-g-AKNN algorithm can effectively deal with the transfer of epidemic patients and the relief distribution.

In the future, the following extension issues can be considered. Patient migration and psychological panic should be considered in emergency management. Second, it is necessary to consider the routing and scheduling of delivery vehicles for the patients and the relief supplies. Third, more epidemiological interventions in lockdown areas need to be considered in emergency management.

## Figures and Tables

**Figure 1 ijerph-20-01765-f001:**
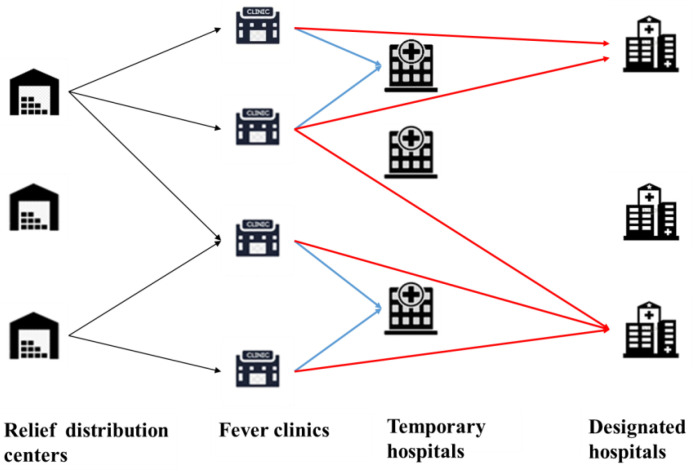
Patient transfer and relief distribution network in lockdown area of epidemic.

**Figure 2 ijerph-20-01765-f002:**
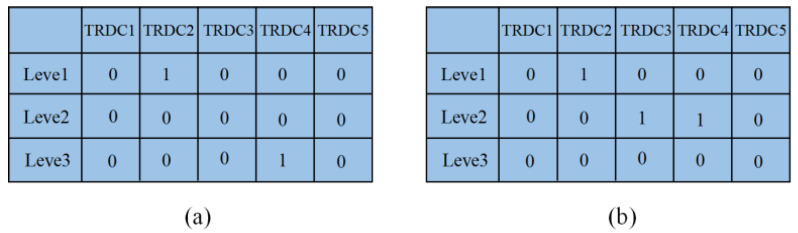
An encoded chromosome for emergency facility location.

**Figure 3 ijerph-20-01765-f003:**
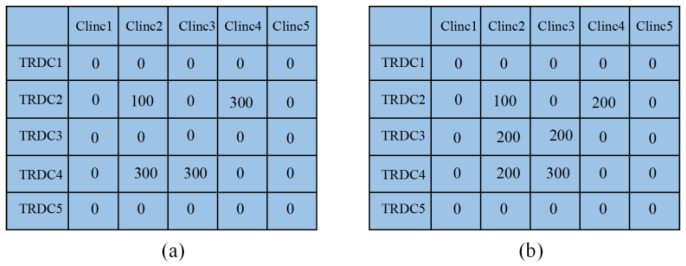
An encoded chromosome for relief planning.

**Figure 4 ijerph-20-01765-f004:**
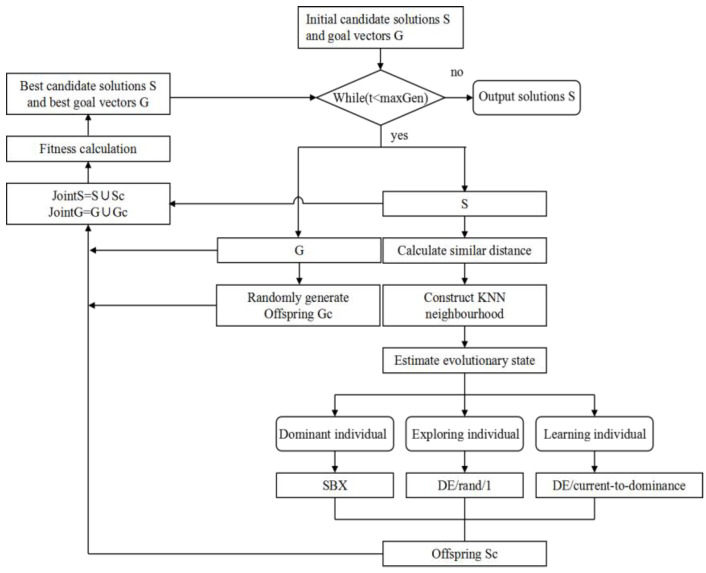
PICEA-g-AKNN framework.

**Figure 5 ijerph-20-01765-f005:**
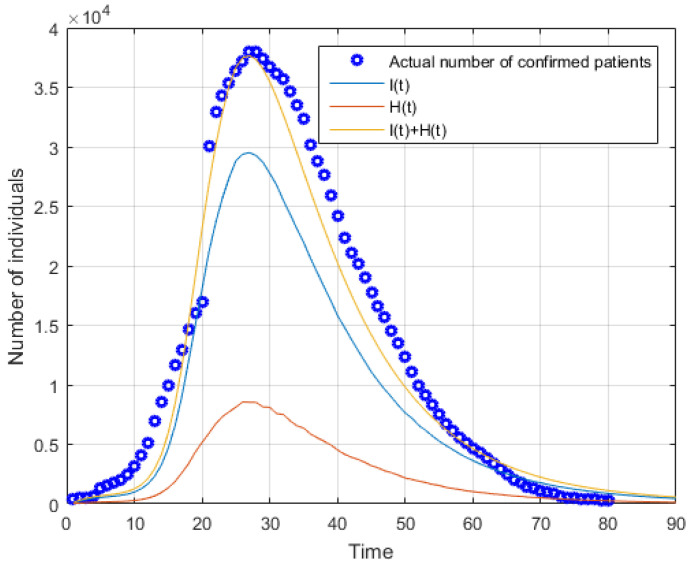
Theoretical estimates of infected patients and the critical patients by the modified SEIR model.

**Figure 6 ijerph-20-01765-f006:**
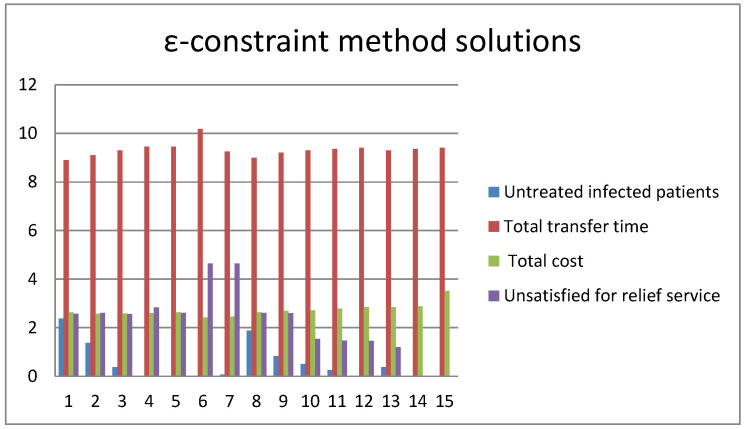
Results for a small test problem by an ε-constraint algorithm.

**Figure 7 ijerph-20-01765-f007:**
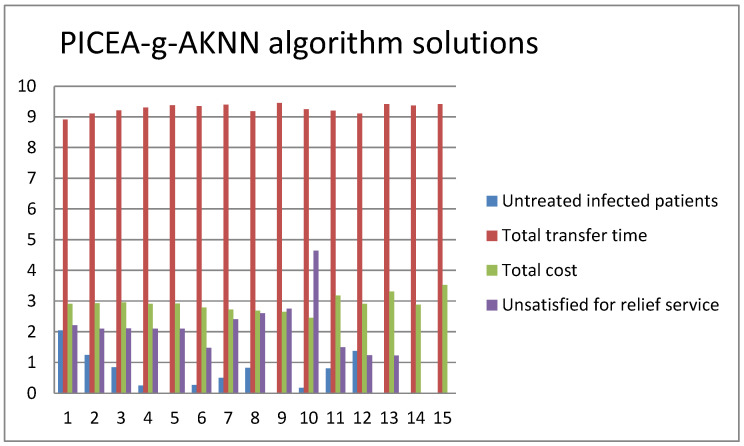
Results for a small test problem by the PICEA-g-AKNN algorithm.

**Figure 8 ijerph-20-01765-f008:**
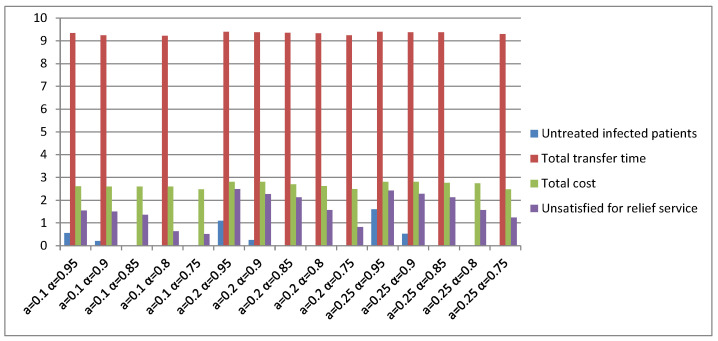
Results in different uncertainty parameters.

**Figure 9 ijerph-20-01765-f009:**
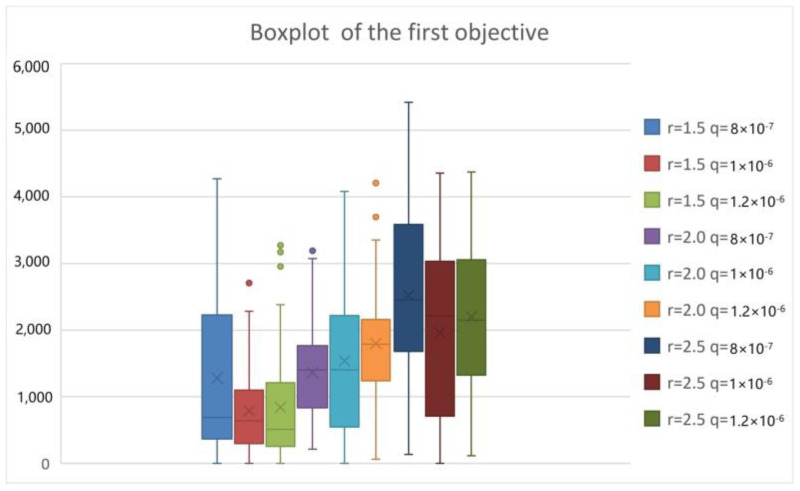
Boxplot chart of the total number of untreated infected patients.

**Figure 10 ijerph-20-01765-f010:**
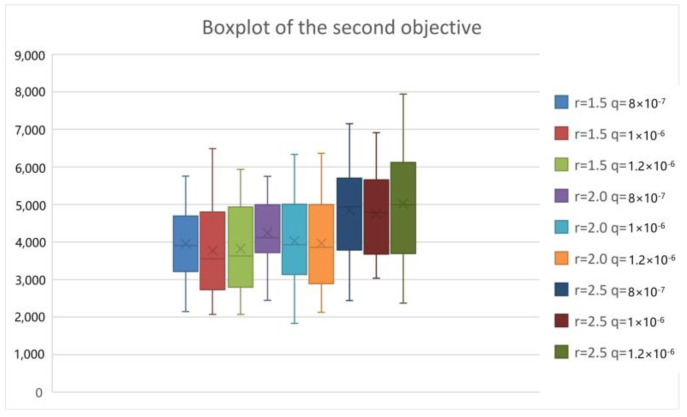
Boxplot chart of the total transfer time.

**Figure 11 ijerph-20-01765-f011:**
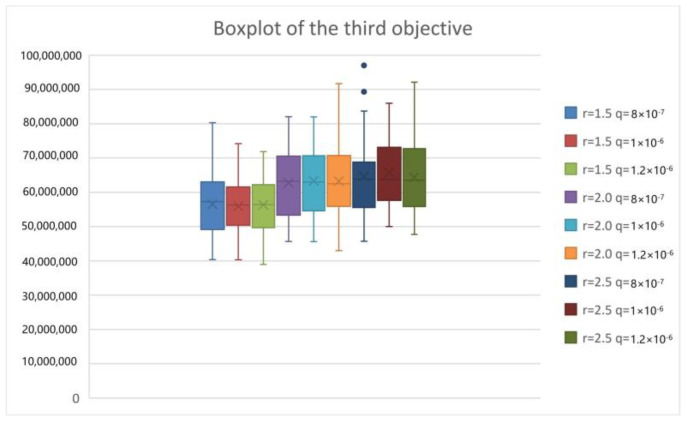
Boxplot chart of the overall cost.

**Figure 12 ijerph-20-01765-f012:**
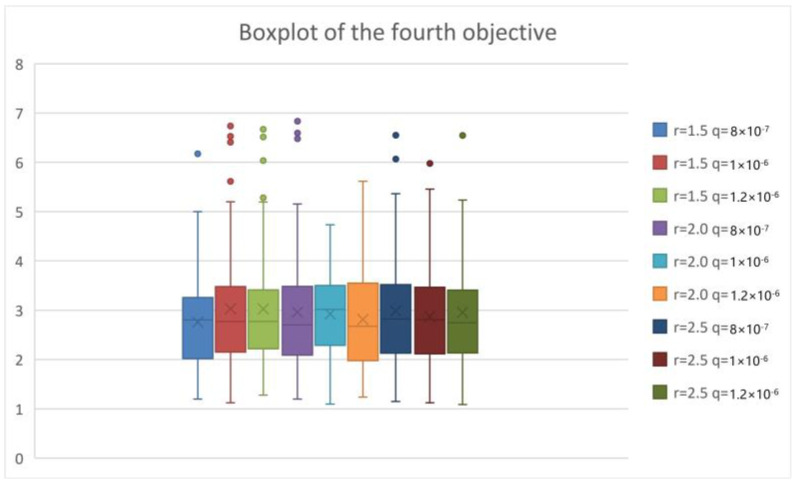
Boxplot chart of criteria of equitable distribution.

**Table 1 ijerph-20-01765-t001:** Parameters in modified SEIR model.

Parameters	r	β	q	α	δ_I_	δ_q_	γ_I_	γ_H_
	2	0.045	1.0 × 10^−6^	2.7 × 10^−4^	0.13	0.13	0.007	0.014

**Table 2 ijerph-20-01765-t002:** Initial values of affected areas.

Affected Areas	1	2	3	4	5	6	7	8	9	10
N	150,005	113,553	175,424	226,502	200,740	164,807	187,551	159,922	210,337	209,542
E(0)	125	131	320	225	477	205	320	103	98	155
I(0)	5	3	9	8	5	7	9	4	7	3
R(0)	37	41	33	36	40	23	35	20	15	32
H(0)	1	3	5	4	2	5	2	3	0	1
S_I_(0)	219	178	409	291	611	321	400	184	129	233
E_I_(0)	139	171	230	180	320	189	306	99	102	174

**Table 3 ijerph-20-01765-t003:** Predicted information of Disaster Area 1.

dI(t)/dH(t)	Period						
Scenario	1	2	3	4	5	6	7
1	15/4	28/8	38/11	46/13	51/15	55/16	57/17
2	15/5	31/10	45/14	56/18	64/21	70/23	77/25
3	15/4	30/9	43/13	54/16	61/18	67/19	70/20
4	25/7	47/14	65/18	78/22	88/25	94/27	98/29

**Table 4 ijerph-20-01765-t004:** Transportation cost and time distance.

$c_{nm}/t_{nm}$	Fever Clinic										
		1	2	3	4	5	6	7	8	9	10
TRDC	1	5/1.2	5/1.1	6/2.0	7/1.6	3/1.1	6/1.5	4/2.2	7/3.0	9/2.5	4/1.3
	2	7/2.0	4/2.2	5/2.3	3/1.5	2/1.0	5/1.3	5/1.3	6/1.5	4/1.8	6/2.4
	3	4/1.5	5/1.7	3/1.0	5/1.5	6/2.0	3/1.1	7/2.6	4/1.6	6/1.7	7/1.9
TH	1	6/1.6	7/1.9	5/1.5	4/1.7	6/2.4	5/2.2	8/2.5	3/1.0	5/1.1	7/1.4
	2	7/2.3	6/2.2	6/3.0	3/1.2	4/1.2	6/1.8	5/2.0	7/2.0	5/1.5	6/1.4
	3	5/1.5	8/2.0	7/2.2	8/2.7	4/2.0	6/1.8	7/2.1	5/1.8	9/2.9	7/3.0
DH	1	5/1.2	6/2.0	3/1.4	6/1.8	4/1.1	7/1.5	5/1.1	8/2.4	3/1.1	8/2.0
	2	7/1.9	5/1.5	7/2.6	6/2.5	5/1.4	8/2.0	8/3.2	6/1.5	3/2.0	7/2.3
	3	5/2.4	6/2.1	4/1.5	6/1.9	7/2.0	3/1.1	6/1.6	9/2.0	5/1.8	4/1.1

**Table 5 ijerph-20-01765-t005:** Hospital parameters.

		Capacity	Fixed Cost
TH	1	2000	30,000
	2	3000	40,000
	3	3000	44,000
DH	1	900	30,000
	2	1000	50,000
	3	800	40,000

**Table 6 ijerph-20-01765-t006:** TRDC parameters.

		Fixed Cost	Variable Cost
TRDC	1	30,000	1.2
	2	25,000	1.5
	3	20,000	1.7

**Table 7 ijerph-20-01765-t007:** Relief commodities sent from established TRDCs to fever clinics.

Scenario	TRDC	Fever Clinic									
		1	2	3	4	5	6	7	8	9	10
1	1	336		29			15	837			460
	2	35	456	652	521	931	513		279	274	
	3										
2	1	263	51	46	18	69	41	713	57	26	393
	2	160	380	630	527	820	495	67	252	259	71
	3										
3	1	347	36	77		32	72	620	38		409
	2	56	390	710	499	861	441	181	237	260	24
	3										
4	1	403	17		21		114	666	94	19	393
	2	102	389	766	559	828	417	58	192	258	83
	3										

**Table 8 ijerph-20-01765-t008:** Infected people transferred from each fever clinic to established THs.

Scenario	TH	Fever Clinic									
		1	2	3	4	5	6	7	8	9	10
1	1	311	377	559					240	234	382
	2				433	760	441	469			
	3										
2	1	382	398	594					288	263	
	2				507	829	496	719			429
	3										
3	1	362	390	756					248	244	
	2				470	791	478	714			394
	3										
4	1		418	781					295	274	
	2				541	866	266	752			435
	3	523					283				

**Table 9 ijerph-20-01765-t009:** Critical patients transferred from each fever clinic to established DHs.

Scenario	DH	Fever Clinic									
		1	2	3	4	5	6	7	8	9	10
1	1	98	80	167	44	229		207		75	
	2										
	3		38	4	91		138		76		119
2	1	134		212		251		221		82	
	2										
	3		125	19	155		155		88		134
3	1	115	12	182	54	240		218		79	
	2										
	3		112		90		147		79		124
4	1	161		165		262		227		85	
	2		131						90		
	3			73	195		168				135

**Table 10 ijerph-20-01765-t010:** Parameters of the test instances.

Instance	|I|	|C|	|J|	|H|	|T|	|S|
T-4-20-4-4-7-4	4	20	4	4	7	4
T-4-20-4-4-7-8	4	20	4	4	7	8
T-4-20-4-4-7-12	4	20	4	4	7	12
T-4-20-4-4-14-4	4	20	4	4	14	4
T-5-20-4-4-14-8	4	20	4	4	14	8
T-5-20-4-4-14-12	4	20	4	4	14	12
T-5-30-5-5-7-4	5	30	5	5	7	4
T-5-30-5-5-7-8	5	30	5	5	7	8
T-5-30-5-5-7-12	5	30	5	5	7	12
T-5-30-5-5-14-4	5	30	5	5	14	4
T-5-30-5-5-14-8	5	30	5	5	14	8
T-5-30-5-5-14-12	5	30	5	5	14	12
T-8-50-8-8-7-4	8	50	8	8	7	4
T-8-50-8-8-7-8	8	50	8	8	7	8
T-8-50-8-8-7-12	8	50	8	8	7	12
T-8-50-8-8-14-4	8	50	8	8	14	4
T-8-50-8-8-14-8	8	50	8	8	14	8
T-8-50-8-8-14-12	8	50	8	8	14	12

**Table 11 ijerph-20-01765-t011:** Initial values of the starting time of the disaster areas.

Initial Values	N	E(0)	I(0)	R(0)	H(0)	Sq(0)	Eq(0)
	[15,000~30,000]	[100~200]	[0~10]	[0~10]	[0~10]	[100~200]	[100~200]

**Table 12 ijerph-20-01765-t012:** Algorithm parameters.

Parameters	PICEA-g-ANK	PICEA-g	MOEA/D	NSGA-II
Maximum generations maxGen Population size N Number of goal vectors Ng Probability pc Probability pm	5000 100 100 0.8 0.2	5000 100 100 0.8 0.2	5000 100 - 0.8 0.2	5000 100 - 0.8 0.2
Neighborhood size T	N/20-N/5	-	20	-

**Table 13 ijerph-20-01765-t013:** Results of the numerical experiments (A: PICEA-g-AKNN, B: PICEA-g, C: MOEA/D, D: NSGA-II).

Instance	$C\left(A,B\right)$	$C\left(B,A\right)$	$C\left(A,C\right)$	$C\left(C,A\right)$	$C\left(A,D\right)$	$C\left(D,A\right)$
T-4-20-4-4-7-4	0.213	**0.253**	**0.202**	0.183	**0.212**	0.013
T-4-20-4-4-7-8	**0.266**	0.135	0.214	**0.288**	**0.302**	0.044
T-4-20-4-4-7-12	**0.184**	0.117	**0.25**	0	**0.360**	0.161
T-4-20-4-4-14-4	0.20	**0.304**	**0.202**	0.104	0.105	**0.255**
T-5-20-4-4-14-8	**0.496**	0.121	**0.24**	0.166	**0.406**	0
T-5-20-4-4-14-12	**0.211**	0.058	0.057	**0.204**	**0.237**	0.187
T-5-30-5-5-7-4	**0.419**	0.013	**0.358**	0.072	0.023	**0.303**
T-5-30-5-5-7-8	0.168	**0.263**	0.072	0.227	0.042	0.164
T-5-30-5-5-7-12	0.283	0.105	**0.299**	0.056	**0.183**	0.133
T-5-30-5-5-14-4	**0.218**	0.152	**0.195**	0.012	0.153	**0.389**
T-5-30-5-5-14-8	**0.285**	0.151	0.158	**0.221**	**0.248**	0.242
T-5-30-5-5-14-12	**0.480**	0	**0.184**	0.118	0.223	**0.294**
T-8-50-8-8-7-4	**0.515**	0.124	**0.231**	0.132	0.171	**0.40**
T-8-50-8-8-7-8	**0.317**	0.15	0.147	**0.294**	**0.297**	0.145
T-8-50-8-8-7-12	**0.275**	0.164	0	**0.412**	0.194	**0.290**
T-8-50-8-8-14-4	**0.360**	0.218	**0.159**	0.113	**0.315**	0.047
T-8-50-8-8-14-8	**0.581**	0	0.125	**0.177**	**0.307**	0.031
T-8-50-8-8-14-12	**0.338**	0.081	**0.149**	0.033	**0.238**	0.155
Average	**0.3629**	0.1506	**0.2026**	0.1758	**0.2510**	0.2031

## Data Availability

The datasets used and/or analyzed during the current study are available from the corresponding author upon reasonable request.

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
