# Peer review of "Two-Stage Multi-Objective Stochastic Model on Patient Transfer and Relief Distribution in Lockdown Area of COVID-19"

_ijerph, 2023, doi:10.3390/ijerph20031765_

Round 1
Reviewer 1 Report
1. Could I ask for an indication of how the model was validated?
2. The wording in the "6. Conclusion and Future Studies" section is at a low level of detail. The first paragraph is an overview reminder of the content of the work, rather than a reference to the main results of the work obtained.
Reviewer 2 Report
The Two-Stage Multi-Objective Stochastic Model proposed in this paper does not express the innovation value in the main text. The background description is messy and the value of the study cannot be judged. I suggest the author further comb the article to highlight the contribution of the article.
1. Current knowledge gap should be explicitly stated in Introduction.
2. Breakthrough of this study is unclear. It should be emphasized in Introduction.
3. Formula and Figure quality should be improved.
4. Impacts to the community should be stated at the end of the manuscript.
Round 2
Reviewer 2 Report
I think the manuscript has been improved after revision. However, we believe that there is room for improvement in the following aspects of the article:
1、When presenting multiple charts, the resulting description does not say which one to refer to. For example Table 3, 4, 5, 6. Figures 9-12. How does the numerical value of the graph illustrate the superiority of the method?
2、The conclusion is that the proposed method is superior to other methods. But it is not clear in what way it is better than other methods. What are the advantages?
